# LoMime: Query-Efficient Membership Inference using Model Extraction in Label-Only Settings

## Abstract

Membership inference attacks (MIAs) threaten the privacy of machine learning models by revealing whether a specific data point was used during training. Existing MIAs often rely on impractical assumptions—such as access to public datasets, shadow models, confidence scores, or training data distribution knowledge—making them vulnerable to defenses like confidence masking and adversarial regularization. Label-only MIAs, even under strict constraints suffer from high query requirements per sample. We propose a cost-effective label-only MIA framework based on transferability and model extraction. By querying the target model $M$ using active sampling, perturbation-based selection, and synthetic data, we extract a functionally similar surrogate $S$ on which membership inference is performed. This shifts query overhead to a one-time extraction phase, eliminating repeated queries to $M$. Operating under strict black-box constraints, our method matches the performance of state-of-the-art label-only MIAs while significantly reducing query costs. On benchmarks including Purchase, Location, and Texas Hospital, we show that a query budget equivalent to testing $\approx 1\%$ of training samples suffices to extract $S$ and achieve membership inference accuracy within $\pm 1\%$ of $M$. We also evaluate the effectiveness of standard defenses (e.g., DP-SGD, regularization) proposed for label-only MIAs against our attack.

## 1 Introduction

The widespread deployment of machine learning (ML) models in sensitive domains (e.g., healthcare (Guerra-Manzanares et al., 2023), finance (Grigoriadis et al., 2023)) raises critical concerns about privacy and model security. These models are often trained on data containing personal medical records, financial transactions, or behavioral patterns. When exposed via public APIs, they can become prime targets for adversaries seeking to exploit privacy vulnerabilities through black-box interactions. A prominent threat in this context is the membership inference attack (MIA), where the attacker aims to determine whether a specific sample was part of a model's training set (Shokri et al., 2017). Even without direct data reconstruction, such inferences can lead to harmful disclosures, such as a patient's participation in a disease-specific clinical trial.

**Related work in membership inference attacks.** Early MIAs (Shokri et al., 2017; Salem et al., 2019; Pyrgelis et al., 2018; Truex et al., 2021; Hayes et al., 2019; Hilprecht et al., 2019; Song et al., 2019; Sablayrolles et al., 2019; Long et al., 2020; Li et al., 2021; Hui et al., 2021; Nasr et al., 2019; Jia et al., 2019; Yang et al., 2020) consider white-box or confidence-based settings, assuming access to confidence scores, auxiliary datasets, or shadow models. These assumptions are often unrealistic in practice and can be mitigated by defenses such as confidence masking (Shokri et al., 2017; Salem et al., 2019; Truex et al., 2021; Jia et al., 2019; Yang et al., 2020), adversarial regularization (Nasr et al., 2019) or generalization enhancement (Shokri et al., 2017; Salem et al., 2019; Truex et al., 2021; Abadi et al., 2016; Srivastava et al., 2014). Recent MIA studies focus on various research directions such as conducting attacks with lower costs (Zarifzadeh et al., 2024), designing attacks against large language models (Duan et al., 2024; Mireshghallah et al., 2022; Mattern et al., 2023), utilizing AI explainability (Liu et al., 2024), using MIAs for auditing models (Ye et al., 2022), proposing attacks with high accuracy in low FPR-regimes (Carlini et al., 2022), or developing attacks under label-only settings (Choquette-Choo et al., 2021; Li & Zhang, 2021). In this work, we focus on the restrictive and

realistic label-only MIAs. Yeom et al. (2018) introduced a naive baseline for label-only membership inferences such that correctly labeled samples are considered as members. Choquette-Choo et al. (2021) introduced a method that infers membership by evaluating the robustness of predicted labels under input perturbations, showing that significant leakage is possible even in highly restricted settings. However, their method requires a source model trained on labeled data to calibrate the membership threshold $\tau$. Li & Zhang (2021) proposed a more constrained approach that calibrates $\tau$ on adversarial synthetic samples which removes the need for external data. Although these techniques reduce the assumptions of early MIAs, they remain highly query-intensive: determining membership for a single sample may still require thousands of queries, and threshold calibration adds further query cost, limiting the scalability of label-only MIAs particularly on large membership datasets.

**Related work in model extraction attacks.** The constraints in label-only MIAs (Choquette-Choo et al., 2021; Li & Zhang, 2021) can be addressed using another privacy attack in the literature. Model extraction attacks (Tramèr et al., 2016; Orekondy et al., 2019; Juuti et al., 2019; Papernot et al., 2017; Jagielski et al., 2020; Truong et al., 2021; Krishna et al., 2020; Karmakar & Basu, 2023) aim to replicate the functionality of a target model by strategically querying it and training a surrogate model to approximate its decision boundaries. These attacks pose a serious threat to ML models deployed online, rendering them vulnerable to extraction and theft. Depending on the available outputs—probabilities, logits, or hard labels—attackers adapt their strategies to recover models with varying fidelity. Extraction is hardest when restricted to hard-label outputs, yet remains attainable.

**The proposed framework.** In this paper, we propose a new attack vector that integrates label-only model extraction with membership inference, designed to operate under the most restrictive conditions. The attacker is assumed to have no access to public datasets, shadow datasets from the target distribution, or knowledge of the model architecture. Extraction is initialized with synthetic data and, at most, a minimal auxiliary dataset containing a few unrelated samples per class. Rather than querying the target model $M$ for each membership decision, the attacker first extract a surrogate model $S$ using a query-efficient sampling strategy and then perform membership inference offline. This is possible thanks to transferability properties reported in prior work (Papernot et al., 2016; Liu et al., 2017; Naseer et al., 2019; Demontis et al., 2019) in which the surrogate model $S$ not only replicates the target's predictions but also retains its membership leakage behavior. This also concentrates the query cost of membership inference into a single extraction phase. While the extraction phase may require more queries upfront than per-sample inference on a small set, it becomes substantially more efficient for large-scale membership inference.

Our framework integrates two complementary techniques for extraction: active-learning–based, query-efficient sampling (cf. MARICH (Karmakar & Basu, 2023)) and adversarial synthetic sample generation with perturbation (cf. AUTOLYCUS (Oksuz et al., 2024)). Neither technique alone suffices—the former presumes access to public data, the latter relies on explanation-based leakage—so we adapt and combine them for a stricter label-only setting in which neither assumption holds. In this setting, synthetic data supplies the query pool that active sampling iteratively refines, yielding a self-contained, data-free extraction strategy consistent with black-box conditions. In the second stage, we apply unsupervised, label-only MIAs to $S$, reflecting scenarios where adversaries have limited or no access to the surrogate's training data yet still wish to conduct offline inference. Moreover, label-only attacks also provide a lightweight alternative to confidence-vector MIAs, which typically require training numerous shadow models (e.g., LiRA (Carlini et al., 2022)) and incur substantial computational and memory overhead. Given sufficient queries, label-only methods approach the effectiveness of confidence-vector attacks, making them a practical choice within our framework.

The rest of this paper is organized as follows. Section 2 describes our system/threat model. Section 3 presents our proposed framework. Section 4 details experimental results. Section 5 evaluates the impact of defenses. Section 6 concludes the paper and suggests potential future directions.

## 2 SYSTEM AND THREAT MODEL

We consider a deployed black-box classifier $M : \mathcal{X} \to \mathcal{Y}$ trained on an unknown dataset $D_M$ as the target model $M$. The attacker trains a surrogate $S$ using only oracle access to $M$ that returns **hard labels** $y_i = M(x_i)$ for inputs $x_i \in \mathcal{X}$. A finite query budget $q$ limits total interaction with $M$ during extraction and all subsequent computations on $S$ are offline and unconstrained by $q$. The attacker has no public or shadow dataset aligned with $D_M$ and no prior knowledge of $M$'s architecture,

hyperparameters, or defenses. At most, a minimal off-distribution auxiliary set $D_A$ (e.g., 1–2 samples per class) may be available to seed synthetic generation. Inputs must satisfy domain constraints (e.g., feature ranges/types, sparsity patterns), and explanations or other side channels are not provided. Defenses such as dropout, $\ell_2$ regularization, or DP-SGD may be present but are unknown to the attacker and treated as part of the environment.

In the extraction phase, the attacker's objective is to learn a surrogate $S$ that closely agrees with $M$, measured by label-agreement fidelity $F_M^S = \Pr_{x \sim \mathcal{D}_N}[S(x) = M(x)]$ on a neutral test set $D_N$ under budget $q$. In the membership inference phase, given an evaluation set $D_{\text{mem}} = \{x_i\}$, the objective is to infer membership bits $\hat{m}_i \in \{0, 1\}$ which indicates whether each $x_i \in D_M$. Inference performance is reported via accuracy and AUC when ground truth is available. A key property of this model is amortization of query cost: all queries to $M$ occur once during the extraction phase, after which membership inference is performed offline on $S$ without further access to $M$. Membership decisions are derived from a scalar robustness/distance-based score computed on $S$ and a single threshold $\tau$ calibrated without querying $M$ (e.g., using synthetic data). The target model excludes access to confidences/logits/explanations, white-box or gradient interfaces, curated public/shadow datasets aligned with $D_M$, poisoning/backdoor manipulations, and traffic/side-channel leakage. Success is demonstrated by achieving high $F_M^S$ and competitive test performance under budget $q$, and by attaining strong membership accuracy/AUC on $D_{\text{mem}}$—ideally within a small margin of attacks run directly against $M$—despite operating under these stricter label-only constraints.

## 3 METHODOLOGY

The first stage of our framework is the model extraction. Since the target model $M$ only provides labels, we require a technique that is both query-efficient and capable of producing surrogate models that are equivalent to the target model in terms of their predictions. In label-only scenarios, the extraction technique we adopt is MARICH (Karmakar & Basu, 2023), a query-efficient, label-only, multi-stage extraction method based on active-learning and, primarily focused on stealing image classifiers by leveraging public datasets as auxiliary resources. These public datasets may or may not be derived from the same distribution as the target model's training data $D_M$. These datasets are utilized in optimized sample selection for the active learning of the surrogates. While MARICH is tailored for image data and, to a degree, text data, applying similar techniques to tabular datasets presents additional challenges due to the absence of spatial relationships and the reliance solely on feature-level attributions. Hence, we incorporate a synthetic data generation scheme from another label-only extraction framework, AUTOLYCUS (Oksuz et al., 2024) to address these challenges and represent a more realistic scenario where public datasets are not available. AUTOLYCUS generates synthetic datasets through dynamic augmentation and perturbation, which are used in optimal sample selection. AUTOLYCUS optimizes data generation by applying perturbations only to the most important $k$ features, as identified via AI explanations (XAI). Since we consider a scenario where XAI is unavailable, we only adopt its probabilistic perturbation strategy. Specifically, our attack combines the synthetic data generation of AUTOLYCUS with MARICH's active learning methodology to construct surrogates $S$. Technical overview of our attack is provided below.

### 3.1 MODEL EXTRACTION ATTACK

In active learning, the extraction process can framed as an optimization problem aimed at maximizing mutual information and label agreement between $M$ and $S$. In each $t^{\text{th}}$ iteration of active learning, let $D_Q^t$ denote the query dataset used for optimization, obtained by augmenting and perturbing $D_A$. The optimization objective is defined in Equation 1, where $I(\cdot)$ represents mutual information and $q^t$ denotes the query budget of $t^{\text{th}}$ iteration. The goal is to reduce the mismatch between the prediction distributions while enhancing the informativeness of the surrogate model in each iteration.

$$\max \sum I\big(\Pr(M(Q)) \parallel \Pr(S(Q))\big), \quad Q \sim D_Q^t \tag{1}$$

The extraction process in our framework consists of five iterative steps: aggregating and perturbing $D_A$ using AUTOLYCUS, followed by entropy sampling, entropy gradient sampling, and loss-based sampling techniques from active learning, as implemented in MARICH, and finally re-training $S$ with the newly acquired samples $D_S^t$.

The extraction begins by training an initial surrogate model $S^0$ using $D_A$ labeled by $M$. For any iteration $t$, $D_A$ is augmented by a factor of $\alpha$, which represents the augmentation factor, creating a query dataset $D_Q^t$ where $D_A$ is repeated $\alpha$ times, with $|D_Q| = |D_A| * \alpha$. Then, each sample in $D_Q^t$ is assigned a unique perturbation mask. This mask alters the features with a probability $\rho$. For binary features, the mask applies Bernoulli noise to flip the features, while for continuous features, the mask applies noise based on the standard deviation $\sigma_j$ of each feature or a user-set arbitrary noise value, while ensuring that the feature remains within its range $R_j$. Once each sample in $D_Q^t$ is ensured to be unique and not part of the existing surrogate training set ($D_S^{t-1}$), $D_Q^t$ is sent for entropy sampling.

In the entropy sampling stage, a subset $Q_{\text{entropy}}^t$ of size $|Q_{\text{entropy}}^t| = B$ is selected from $D_Q^t$ by maximizing the entropy of predictions from the surrogate model $S^{t-1}$ as in Equation 2, where $H(\cdot)$ denotes the entropy function. The subset $Q_{\text{entropy}}^t \subseteq D_Q^t$ denotes where $S^{t-1}$ is least certain.

$$Q_{\text{entropy}}^t = \arg \max_{Q \subseteq D_Q^t, |Q| = B} H\big(S^{t-1}(Q)\big), \tag{2}$$

In the entropy gradient sampling stage, $Q_{\text{entropy}}^t$ is further refined by clustering the entropy gradients $\nabla_Q H(S^{t-1}(Q))$ of the selected queries using $k$-means clustering. The number of clusters $k$ typically corresponds to the number of possible labels $n_c$. The objective is to ensure that a sufficiently diverse subset of selected queries $Q_{\text{entropy}}^t$ from each label is used during the extraction process. From the clustered queries, the most diverse subset $Q_{\text{grad}}^t$, of size $\gamma_1 B$, is selected to maximize input-space variability coverage as formulated in Equation 3, where $C$ represents the cluster centers. This stage enhances the robustness and diversity of the extraction process by ensuring balanced label representation and broad input-space variability.

$$Q_{\text{grad}}^t = \arg \min_{Q \subseteq Q_{\text{entropy}}^t, |Q| = \gamma_1 B} \sum_{x_i \in Q} \sum_{c \in C} \|\nabla_q H(S^{t-1}(x_i)) - c\|^2, \tag{3}$$

In the loss sampling stage, $Q_{\text{grad}}^t$ is further refined by selecting $\gamma_1 \gamma_2 B$ queries, denoted as $Q_{\text{loss}}^t$. These are the samples from $Q_{\text{grad}}^t$ that are closest to the surrogate training samples, with the greatest loss between the target model $M$ and the surrogate model $S^{t-1}$. The formulation of loss sampling is provided in Equation 4 For each $x \in D_S^{t-1}$, we compute the cross-entropy loss of $S^{t-1}$ using the label $y$ provided by $M$ as $L(x) = -\log p_{S^{t-1}}(y \mid x)$. Top $k$ samples from $D_S^{t-1}$ with the highest loss are selected as $D_{S\_topk\_loss}^{t-1}$. Then, $\gamma_1 \gamma_2 B$ samples from $Q_{\text{grad}}^t$ closest to $D_{S\_topk\_loss}^{t-1}$ are selected as $Q_{\text{loss}}^t$. Once $Q_{\text{loss}}^t$ is selected, it is sent to $M$ to obtain predicted labels $Y_{\text{loss}}^t = M(Q_{\text{loss}}^t)$. These labeled samples are added to the training set as $D_S^t = D_S^{t-1} \cup Q_{\text{loss}}^t$ and $Y_S^t = Y_S^{t-1} \cup Y_{\text{loss}}^t$.

$$Q_{\text{loss}}^t = \arg \min_{Q \subseteq Q_{\text{grad}}^t, |Q| = \gamma_1 \gamma_2 B} \sum_{x_i \in Q} \sum_{s \in D_{S\_topk\_loss}^{t-1}} \|x_i - s\|^2. \tag{4}$$

Older surrogate model $S^{t-1}$ is updated to $S^t$ by being trained on the newer extended dataset as $S^t = \text{train}(D_S^t, Y_S^t)$, using standard optimization techniques such as stochastic gradient descent or Adam (Kingma & Ba, 2015). This query selection and model training process is repeated iteratively until the total query budget $q$ is exhausted or the desired level of fidelity $F_M^S$ between the target and the surrogate model is achieved. At the end of the process, the final surrogate model $S$ (or $S_{\text{final}}$) approximates $M$ in terms of predictive distribution. This approach ensures high fidelity, informativeness, and diversity while maintaining query efficiency.

## 3.2 MEMBERSHIP INFERENCE ATTACK

Extracting functionally equivalent surrogate model $S$ from the target model $M$ allows the attacker to perform membership inference attacks (MIAs) on a dataset $D_{\text{mem}}$ (whose membership is aiming to determine) offline, circumventing the high query costs per sample associated with label-only attacks. This also eliminates unrealistic assumptions, such as the use of shadow datasets and an excessive number of online adversarial membership queries, which are typically required by both traditional and label-only MIAs. Label-only MIAs achieve performance comparable to traditional MIAs when query budgets are not constrained (Choquette-Choo et al., 2021). To simulate a scenario where $S$ may originate from an external resource or be deployed in an open repository with no access to $D_S$,

and to account for the assumption that no data from the training distribution of the target model $M$ is available, we conduct an unsupervised label-only MIA on $S$ in the second stage of our attack.

$$d_{\text{boundary}}(x_i) = \min_{\delta_i \in \mathbb{R}^d} \|x_i + \delta_i\|_2 \quad \text{subject to} \quad S(x_i + \delta_i) \neq S(x_i). \tag{5}$$

For each sample $x_i \in D_{\text{mem}}$, the unsupervised label-only membership inference attack is based solely on the hard-label prediction $\hat{y}_i = S(x_i)$. The goal is to infer whether the sample $x_i$ was part of the training set of the target model $M$ based on its proximity to the decision boundary of the surrogate model $S$. To perform this inference, we calculate the decision boundary distance $d_{\text{boundary}}(x_i)$ for each sample $x_i \in D_{\text{mem}}$ using Equation 5.

$$\hat{m}_i = \begin{cases} 1 & \text{if } d_{\text{boundary}}(x_i) \geq \tau \\ 0 & \text{if } d_{\text{boundary}}(x_i) < \tau, \end{cases} \tag{6}$$

This distance quantifies how far $x_i$ is from the nearest decision boundary of $S$. Using the calibrated threshold $\tau$, we classify $x_i$ as a member if $d(x_i) \geq \tau$ and as a non-member otherwise (see Eq. 6), where $\hat{m}_i$ denotes the predicted membership label. The intuition is that training members tend to lie farther from the model's decision boundary due to overfitting.

$$d_{\text{boundary}}(x_i) = \min_{\delta_i \in \mathbb{R}^d} \|\delta_i\|_2 \quad \text{subject to} \quad S(x_i + \delta_i) \neq S(x_i). \tag{7}$$

MIA begins with the calibration of the decision boundary distance threshold $\tau$ on $S$. In order to calibrate $\tau$, we first generate a set of random samples $X_{\text{random}} = \{x_i \in \mathbb{R}^d \setminus D_S\}$, where each sample $x_i$ is drawn uniformly random from the feature space of the target model's training data. These synthetic samples serve as surrogate inputs for calibrating the decision boundary distance. For each generated sample $x_i$, the goal is to determine the minimal adversarial perturbation $\delta_i \in \mathbb{R}^d$ needed to change model's prediction. This is achieved by solving the optimization problem in Equation 7.

$$\tau = \max_{x_i \in X_{\text{random}}} d_{\text{boundary}}(x_i). \tag{8}$$

After computing the decision-boundary distance $d_{\text{boundary}}(x_i)$ for each sample $x_i$, we set the threshold $\tau$ to the maximum of these distances (Eq. 8). This threshold serves as the cutoff at which the model's classification is expected to change and is used to determine membership for other samples. The scalar threshold can be generalized to class-specific thresholds $T = \{\tau_1, \tau_2, \ldots, \tau_{n_c}\}$ when auxiliary information (e.g., decision regions, training or population distributions, or partial training data) is available. However, because decision regions in complex models are highly irregular and nonlinear and the attacker has little or no auxiliary information, we adopt a single global threshold.

### 3.3 PERFORMANCE EVALUATION

We assess the attack along two axes—*extraction* and *membership inference*. For extraction, we report (i) fidelity $F_M^S$ (label agreement between $S$ and $M$), (ii) test accuracy $\text{Acc}(\cdot, D_N)$ of $S$ and $M$, and (iii) the query budget $q$ consumed during extraction. For membership inference, we report (i) accuracy on $D_{\text{mem}}$ at a calibrated threshold and (ii) threshold-free AUC from continuous membership scores.

High $F_M^S$ together with competitive $\text{Acc}(\cdot, D_N)$ at a modest $q$ indicates a successful, cost-effective extraction. Comparing membership metrics between $S$ and $M$ evaluates leakage transfer: close accuracy/AUC implies that $S$ preserves the target's membership signal, whereas a gap suggests incomplete transfer (e.g., due to underfitting during extraction). Since membership inference on $S$ is conducted offline, our approach is advantageous at scale. Once $S$ is extracted, large $D_{\text{mem}}$ can be evaluated without additional queries to $M$.

## 4 EVALUATION

In this section, we describe the datasets, experimental setup, metrics and the obtained results.

### 4.1 DATASETS

We evaluate our framework using three benchmark datasets widely adopted in privacy and security research Shokri et al. (2017): **Location**, **Purchase**, and **Texas Hospital**. The key statistics of these datasets, training configurations, selected perturbation factors, and target model performances are summarized in Table 1. Note that all target models are overfitted and achieve close to perfect training accuracy ($\approx 100\%$) to simulate realistic privacy risks in deployed machine learning systems.

Table 1: Dataset statistics, training configurations, and target model performance.

| Dataset | # of Total Samples ($|\mathbf{D}|$) | # of Features ($\mathbf{n_j}$) | # of Classes ($\mathbf{n_c}$) | # of Training Samples ($|\mathbf{D_M}|$) | # of Auxiliary Samples ($|\mathbf{D_A}|$) | $\rho$ | M's Testing Accuracies |
|---|---|---|---|---|---|---|---|
| Location | 5,010 | 446 | 30 | 1,600 | 150 | 0.10 | $0.6033 \pm 0.0084$ |
| Purchase | 197,324 | 600 | 100 | 10,000 | 1,000 | 0.08 | $0.6489 \pm 0.0031$ |
| Texas Hospital | 67,330 | 6,170 | 100 | 10,000 | 1,000 | 0.005 | $0.4819 \pm 0.0024$ |

## 4.2 EXPERIMENTAL SETUP

In our experiments, both target $M$ and surrogate models $S$ are implemented and trained using **PyTorch**. Following the setup of Shokri et al. (2017), $M$ are configured as feedforward neural networks with a single hidden layer of **128 nodes** and the **tanh** activation function. Models are trained for up to **200 epochs** using the **AdamW** optimizer (Loshchilov & Hutter, 2019), with a learning rate of **0.001** and a weight decay coefficient of $\lambda = \mathbf{1e\text{-}7}$. Batch shuffling is applied at each epoch, with batch sizes of **100** for the Location dataset and **200** for the Purchase and Texas Hospital datasets. To simulate the adversary's limited knowledge of dataset characteristics, augmentation and perturbation parameters used across all datasets during model extraction are set to $k = \mathbf{n_j}$ (uniform binary flipping), $\alpha = \mathbf{4}$, and $\gamma_1 = \gamma_2 = \mathbf{0.5}$. Apart from $\rho$, introduced parameters are architecture agnostic in our framework and primarily regulate per-iteration training-set growth rather than extraction quality. If $\rho$ is too large, perturbations yield unrealistic queries; if too small, queries become uninformative—both cases degrade extraction fidelity. Therefore, $\rho$ needs to be set before the query budget is spent. A set of synthetic samples or the initial $D_A$ can be utilized to configure $\rho$ as $\sigma(|x_i > 0|), x_i \in D_A$ ($\sigma$ is standard deviation). Further details on variability and computational resources used in our experimental setup is in Section C of Appendix.

Membership inference attacks are conducted in a strict **label-only setting**, leveraging the surrogate models to minimize query overhead. We use the **Adversarial Robustness Toolbox (ART)** of IBM (Nicolae et al., 2019) to implement standardized label-only membership inference attacks.

## 4.3 EXPERIMENTS

We assess the effectiveness of our attacks using five key metrics, grouped into two categories: **model extraction metrics** and **membership inference metrics**. The first three—**fidelity of the surrogate model**, **test accuracy** and **the query budget**—primarily evaluate the model extraction stage. Fidelity ($F_M^S$) quantifies the label agreement between $S$ and $M$ over a neutral dataset ($D_N$), which measures the percentage of predictions where $S$ and $M$ produce identical outputs, indicating how accurately $S$ replicates $M$'s decision boundaries. Test accuracy reflects the classification performance of both the surrogate model ($S$) and the target model ($M$) on $D_N$ and its ground-truth labels, providing a standardized evaluation of their generalization capabilities. The query budget determines the number of samples allowed to be classified by $M$. The remaining two metrics—**attack accuracy** and the area under the ROC curve (**AUC**)—are used to assess membership inference effectiveness, based on false positive rates (FPR) and true positive rates (TPR). While each metric primarily supports its corresponding attack component, they also offer insight into the relationship between extraction fidelity and membership leakage.

We further evaluate the model extraction process by measuring how well the surrogate model $S$ replicates the membership inference performance of the target model $M$ in a black-box, label-only setting. We evaluate the attack across surrogate models which have varying query budgets $q$. For the Location dataset, we use $q \in \{1K, 10K, 100K\}$, and for the Purchase and Texas datasets, $q \in \{10K, 100K, 1M\}$. Accordingly, a model denoted as $S_{100K}$ represents a surrogate trained by querying the target model 100,000 times. Our extraction and inference performances across different datasets and $q$ are provided in Table 2, and ROC curves are shown in Figure 1.

Across all datasets, we observe that increasing the query budget leads to consistent and often non-linear improvements in both model fidelity and membership inference performance. On the Location dataset, $S_{1K}$ achieves a fidelity of $0.4969$ and an AUC of $0.5722$, which rise to $0.9081$ and $0.8582$ at $S_{100K}$. ROCs in Figure 1a highlights that $M$s are not fully extracted with $q = 100K$ but membership inference accuracies are in $2 - 3\%$ distance. If $q$ is to be increased by 2-3x, perfect extraction and unlimited inference is a possibility. ROC curve of $S_{10K}$ is closer to its successor $S_{100K}$ more than the

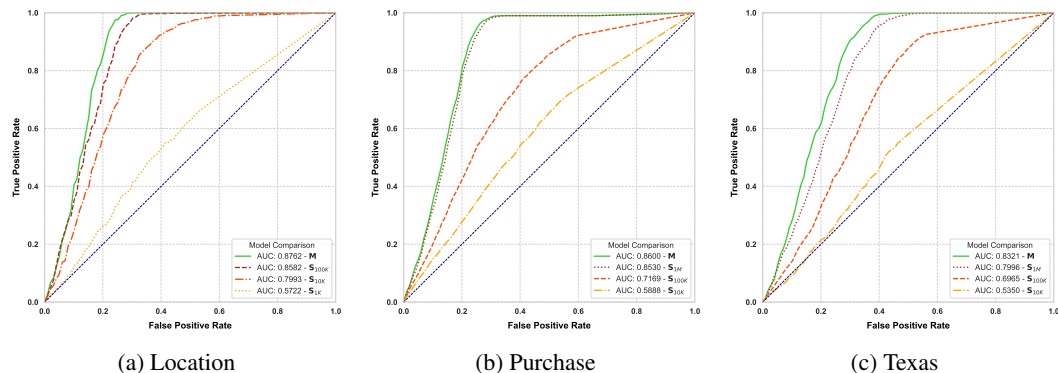

| (a) Location | (b) Purchase | (c) Texas |
|:---:|:---:|:---:|

Figure 1: ROC curves for membership inference attack on surrogate and target models across datasets.

Table 2: Performance summary of privacy attacks under varying query budgets, with model extraction results shown in the upper block and membership inference results in the lower block.

| Dataset | Model Similarities ($F_M^S$) | | | Test Accuracies | | | |
|---|---|---|---|---|---|---|---|
| | $S_{1K}$ | $S_{10K}$ | $S_{100K}$ | $S_{1K}$ | $S_{10K}$ | $S_{100K}$ | $M$ |
| **Location** | $0.4969 \pm 0.0084$ | $0.7619 \pm 0.0080$ | $0.9081 \pm 0.0044$ | $0.4426 \pm 0.0059$ | $0.5584 \pm 0.0090$ | $0.5967 \pm 0.0086$ | $0.6033 \pm 0.0084$ |
| | $S_{10K}$ | $S_{100K}$ | $S_{1M}$ | $S_{10K}$ | $S_{100K}$ | $S_{1M}$ | $M$ |
| **Purchase** | $0.5743 \pm 0.0052$ | $0.7043 \pm 0.0042$ | $0.9620 \pm 0.0033$ | $0.5418 \pm 0.0054$ | $0.6026 \pm 0.0038$ | $0.6486 \pm 0.0031$ | $0.6489 \pm 0.0031$ |
| **Texas** | $0.5329 \pm 0.0049$ | $0.7592 \pm 0.0026$ | $0.8966 \pm 0.0054$ | $0.4229 \pm 0.0044$ | $0.4863 \pm 0.0015$ | $0.4844 \pm 0.0034$ | $0.4819 \pm 0.0024$ |
| Dataset | Attack Accuracies | | | | AUC | | | |
| | $S_{1K}$ | $S_{10K}$ | $S_{100K}$ | $M$ | $S_{1K}$ | $S_{10K}$ | $S_{100K}$ | $M$ |
| **Location** | $0.5873 \pm 0.0311$ | $0.7773 \pm 0.0199$ | $0.8563 \pm 0.0161$ | $0.8783 \pm 0.0138$ | $0.5722 \pm 0.0378$ | $0.7993 \pm 0.0242$ | $0.8582 \pm 0.0274$ | $0.8762 \pm 0.0155$ |
| | $S_{10K}$ | $S_{100K}$ | $S_{1M}$ | $M$ | $S_{10K}$ | $S_{100K}$ | $S_{1M}$ | $M$ |
| **Purchase** | $0.5850 \pm 0.0110$ | $0.6882 \pm 0.0116$ | $0.8495 \pm 0.0091$ | $0.8552 \pm 0.0089$ | $0.5888 \pm 0.0126$ | $0.7169 \pm 0.0158$ | $0.8530 \pm 0.0107$ | $0.8600 \pm 0.0097$ |
| **Texas** | $0.5467 \pm 0.0066$ | $0.6953 \pm 0.0102$ | $0.7823 \pm 0.0137$ | $0.8123 \pm 0.0084$ | $0.5350 \pm 0.0074$ | $0.6965 \pm 0.0084$ | $0.7996 \pm 0.0132$ | $0.8321 \pm 0.0092$ |

($S_{100K}$, $S_{1M}$) pairs of Purchase and Texas datasets. This can be explained with the lesser complexity of $M$ (due to smaller $n_c$, $n_j$ and $|D_M|$) trained on Location dataset.

On the Purchase dataset, the surrogate's fidelity reaches $0.9620$ and AUC reaches $0.8530$ at $S_{1M}$, nearly matching the target model's AUC of $0.8600$. ROCs in Figure 1b highlights that $S_{100K}$ has the least similarity with $M$ among the three datasets and, it is reflected as a sharp decrease in membership inference attack accuracy and AUCs. Despite this, $S_{1M}$ and $M$ have almost identical membership inference performances, congruent to their model similarity. This allows the unlimited number of samples to be inferred and potentially allowing more advanced attacks (e.g., model inversion) to be conducted on $M$. Considering the complexity of $M$ ($n_c$, $n_j$ and $|D_M|$) trained on Purchase dataset, $q = 1M$ is sufficient enough for almost full extraction.

In contrast, the Texas dataset shows a more gradual trajectory. At $S_{1M}$, the surrogate achieves $0.8966$ fidelity and $0.7996$ AUC, still below the target's $0.8321$ AUC. The slower convergence suggests that high-dimensional, sparse, or less structured datasets require more extensive exploration to accurately approximate membership-sensitive regions. Furthermore, comparison between the ROC curves of $S_{10K}$ in Texas and $S_{1K}$ in Location demonstrate that higher $F_M^S$ does not equate a better attack accuracy or AUC in membership inference. This emphasizes that decision boundary alignment in areas critical to high-precision privacy attacks is more difficult to replicate in complex feature spaces especially under low query budget regimes.

Overall, the results demonstrate that high surrogate fidelity and sufficient query budget $q$ are critical for strong membership inference performance, as the models get more complex and the data gets more sparse. On Location, $S_{10K}$ achieves an AUC of $0.7993$ with only $0.7619$ fidelity, suggesting that partial boundary recovery may suffice in lower-dimensional or well-generalized settings. In contrast, for Purchase and Texas, meaningful replication of membership leakage only emerges once fidelity exceeds $0.90$ and $q > 100K$, indicating that more complex or sparse feature spaces require

tighter alignment with the target's decision boundaries. These findings highlight the potential need for further querying or targeted optimization over underrepresented classes or feature regions.

**Cost-Effectiveness and Comparison to Prior Work.**

Label-only membership inference attacks such as the one proposed by Choquette-Choo et al. (2021) achieve strong MIA performance, but they incur significant query costs—approximately 10,000 queries per victim sample. In their reported experiments, they achieve membership accuracies of 0.8920, 0.8740, and 0.8030 on Location, Purchase, and Texas, respectively. However, conducting the attack for 100 victim samples would require 1 million queries in total with their given query budget.

Our approach reallocates query budgets of membership inference to extract a surrogate model $S$ from the target model $M$, which can then be queried offline to perform membership inference on an arbitrary number of samples at no additional cost. This amortization makes our attack significantly more cost-effective when the goal is to evaluate membership across many samples. Once $S$ is sufficiently extracted, inference is performed entirely offline, without incurring further query overhead with adversarial perturbations, and without requiring confidence scores or shadow models.

Our method may not always be the preferable option: if the number of samples to be evaluated is small, direct label-only membership inference—despite its high per-sample cost—may remain the more efficient route. Our approach is more appropriate for adversaries seeking to audit or infer membership for large populations, where the one-time cost of surrogate extraction can be justified.

To initiate model extraction, we used small auxiliary datasets outside of the distribution of $D_M$, typically constrained to 10% of the size of the target model's training set. While full synthetic initialization is also possible, doing so would require more queries and risk generating unrealistic samples in structured domains—hence we maintained minimal auxiliary data as a practical compromise in our experiments.

Overall, our results demonstrate that with a fixed query budget, attackers can trade per-sample overhead for one-time extraction cost, achieving scalable and reusable membership inference. The surrogates not only approximate the prediction function of the target model, but also reproduces its decision boundary vulnerabilities, highlighting a viable and generalizable pathway for privacy leakage in label-only black-box settings.

## 5  COUNTERMEASURES

We evaluate the impact of three defense mechanisms—DP-SGD (Abadi et al., 2016), Dropout (Srivastava et al., 2014), and L2 regularization (Ng, 2004; Krogh & Hertz, 1991) on the performance of our attack. These techniques are selected based on their reported effectiveness against label-only membership inference attacks, as demonstrated by (Choquette-Choo et al., 2021). All models are trained on the same training split of the Location dataset, with only the applied defense varying. The results are summarized in Table 3.

In our experiments, the target model with applied defenses is denoted as $M'$, while the undefended model is denoted as $M$. As expected, the undefended model $M$ exhibits the highest vulnerability, yielding AUC scores of 0.8919 and 0.8964 when attacked directly and through its surrogate model $S$, respectively. These results indicate substantial membership leakage in both the target and surrogate models, confirming that, in the absence of defense mechanisms, our attack remains highly effective.

Among the evaluated defenses, DP-SGD demonstrates the strongest reduction in attack success. Particularly at $\epsilon = 20$, where the privacy guarantee is strongest due to the highest level of noise injection, we observe a substantial drop in AUC on $M'$ (0.7636) and $S$ (0.7351), relative to the undefended model. DP-SGD can be applied with even lower $\epsilon$s. The key takeaway is that the resulting model's fidelity to the original target model $M$ decreases drastically, as evidenced by $F_M^{M'}$ scores. If the target model is not intentionally overfitted for functional fidelity, it offers the best privacy protection and effectively decrease the capabilities of even label-only MIAs.

Dropout, while less disruptive to utility, proves less effective at reducing membership leakage. Even at a high dropout rate ($p = 0.8$), AUC values on $M'$ and $S$ remain elevated, and the surrogate model continues to closely approximate the target. This suggests that Dropout alone is insufficient to

Table 3: Effect of countermeasures on attack and model performance in the Location dataset.

| Defensive Strategy | Setting | $M'$ Metrics | | | | $S$ Metrics | | | | |
|---|---|---|---|---|---|---|---|---|---|---|
| | | $F_M^{M'}$ | Test Acc | Attack Acc | AUC | $F_M^S$ | $F_{M'}^S$ | Test Acc | Attack Acc | AUC |
| **DP-SGD** | $\epsilon = 20$ | 0.6598 | 0.5996 | 0.7367 | **0.7636** | 0.6621 | 0.8541 | 0.6002 | 0.7200 | **0.7351** |
| | $\epsilon = 50$ | 0.7044 | 0.6005 | 0.8233 | **0.8383** | 0.7021 | 0.8605 | 0.6121 | 0.7900 | **0.8178** |
| | $\epsilon = 100$ | 0.7060 | 0.6116 | 0.8433 | **0.8488** | 0.7076 | 0.8612 | 0.6076 | 0.7800 | **0.7916** |
| | $\epsilon = 200$ | 0.7169 | 0.6087 | 0.8400 | **0.8327** | 0.7012 | 0.8500 | 0.5999 | 0.8133 | **0.8305** |
| **Dropout** | $p = 0.2$ | 0.8682 | 0.6189 | 0.8767 | **0.8546** | 0.8480 | 0.9080 | 0.6140 | 0.8567 | **0.8442** |
| | $p = 0.4$ | 0.8548 | 0.6028 | 0.8800 | **0.8674** | 0.8391 | 0.9077 | 0.6040 | 0.8400 | **0.8636** |
| | $p = 0.6$ | 0.8423 | 0.6013 | 0.8700 | **0.8879** | 0.8275 | 0.9157 | 0.5915 | 0.8300 | **0.8448** |
| | $p = 0.8$ | 0.8028 | 0.6128 | 0.8467 | **0.8500** | 0.7932 | 0.9272 | 0.6143 | 0.8200 | **0.8415** |
| **L2 Reg.** | $\lambda = 0.0001$ | 0.8535 | 0.6291 | 0.8667 | **0.8581** | 0.8346 | 0.9160 | 0.6140 | 0.8567 | **0.8548** |
| | $\lambda = 0.0005$ | 0.8378 | 0.6323 | 0.8667 | **0.8532** | 0.8195 | 0.9182 | 0.6236 | 0.8433 | **0.8461** |
| | $\lambda = 0.001$ | 0.8269 | 0.6379 | 0.8467 | **0.8572** | 0.8086 | 0.9186 | 0.6326 | 0.8400 | **0.8614** |
| | $\lambda = 0.005$ | 0.8102 | 0.6660 | 0.8167 | **0.8312** | 0.7967 | 0.9160 | 0.6640 | 0.8200 | **0.8300** |
| **Undefended (M)** | – | – | 0.6078 | 0.8867 | **0.8919** | 0.9128 | – | 0.5999 | 0.8833 | **0.8964** |

significantly hinder our attack pipeline, particularly the membership inference stage. Similarly, L2 regularization by itself exhibits little privacy protection. Increasing the regularization parameter $\lambda$ slightly reduces the extraction fidelity and membership inference success, while improving generalization accuracy. At most $\lambda = 0.005$, we observe at most 6–7% reduction in AUC on both $M'$ and $S$ relative to the undefended case.

Overall, our findings highlight that while DP-SGD provides the most substantial privacy gains and a viable option even against label-only membership inference attacks. L2 regularization does not offer substantial privacy improvements but it increases the test accuracy of $M$. It can be a good defense when merged with DP-SGD for privacy protection and increased utility. Finally dropout, by itself offer little mitigation in privacy and has negligible impact on utility, making it the least impactful of the three proposed countermeasures.

## 6 CONCLUSION AND FUTURE DIRECTIONS

We present a label-only membership inference attack that operates under minimal auxiliary knowledge by first extracting a surrogate model using a query-efficient model extraction strategy. This enables offline inference without further queries to the target. On benchmark tabular datasets, our method matches the performance of prior label-only attacks while requiring significantly fewer queries—achieving comparable accuracy and AUC using a query budget equivalent to inferring membership of only 1% of the target's training data. As the number of evaluated samples increases, our attack gets more cost-effective. We assess the robustness of our approach under standard defenses dropout, $L_2$ regularization, and DP-SGD. Apart from DP-SGD, the privacy protections they offer are very limited, resulting in only minor reductions in membership accuracy and AUC ($\leq 6 - 7\%$). Our findings show that even under the strictest constraints, label-only interfaces remain susceptible to increasingly efficient attacks, while the protection offered by standard defenses continues to diminish. As future work, we aim to improve the efficiency and generality of our method for complex models and high-dimensional data types such as images and sequences.

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

## A DISCLAIMER

In this paper, LLMs (e.g., ChatGPT 5) are used for proofreading and polishing the writing. LLMs are not used for retrieval and discovery (e.g., finding related work), research ideation or any other purposes.

## B COMMONLY USED SYMBOLS AND NOTATIONS

The following table summarizes the commonly used symbols and notations in this paper:

Table 4: Commonly Used Symbols and Notations

| Symbol | Description |
|---|---|
| $\mathbf{M}$ | Target (victim) model |
| $\mathbf{S}$ | Surrogate (a.k.a extracted or stolen) model |
| $\mathbf{D_M}$ | Dataset used to train the target model |
| $\mathbf{D_S}$ | Dataset used to train the surrogate model |
| $\mathbf{D_A}$ | Auxiliary dataset used for training $S^0$ and synthesizing samples |
| $\mathbf{D_Q}$ | Augmented and perturbed query dataset used in active learning for training $S$ |
| $\mathbf{D_N}$ | Neutral dataset used for evaluating model fidelity |
| $\mathbf{D_{mem}}$ | Membership dataset used for evaluating membership inference attack performance |
| $x_i$ | Input sample to the model |
| $y_i$ | Predicted class (label) for the sample $x_i$ |
| $n_j$ | Number of features in the input |
| $n_c$ | Number of output classes |
| $R_j$ | Range of possible values for feature $j$ |
| $R_c$ | Range of possible values for classes (labels) |
| $\tau$ | Decision boundary distance threshold |
| $\mathcal{L}$ | Loss function |
| $\nabla_\theta \mathcal{L}$ | Gradient of the loss with respect to model parameters |
| $F_M^S$ | Fidelity of the surrogate model $S$ relative to the target model $M$ |
| $q$ | Query budget for the adversary |

## C EXPERIMENTAL SETUP - VARIABILITY AND COMPUTATIONAL SETTINGS

In the regular and countermeasure experiments, we ran each experiment multiple times (10+) to ensure result stability and to report means and standard deviations of key metrics given different query budgets $q$. Across different datasets and configurations, we observed consistent trends with minimal variation. Note that, the standard deviations reported are between different runs with different target and surrogate models, data splits. Despite the existing randomness in query generations (model extraction) and randomized samples used for calibrating $\tau$ (membership inference), inner variability of results when the same models (e.g., $M$, $S_{100K}$) are attacked or evaluated with the same splits of the data (e.g., $D_A$, $D_N$) are far lesser than the reported deviations.

All experiments were conducted on an Apple M2 Pro Mac Mini (A2816) with a 10-core CPU, 16-core GPU, 16-core Neural Engine, 16 GB of unified memory, and 1 TB of disk space. All machine learning models were trained using the CPU. The overall compute cost was modest, with each experiment completing in under a few hours. However, it is important to note that increasing the query budgets or the internal parameters of the membership inference attack (e.g., the number of adversarial samples generated) may lead to higher runtime and memory consumption. The total compute usage for the project, including preliminary and discarded trials, remained within the limits of a single-CPU setup and is reproducible on commodity hardware.

