# OpenReview forum: "LoMime: Query-Efficient Membership Inference using Model Extraction in Label-Only Settings"
_ICLR.cc/2026/Conference — Submitted to ICLR 2026_

### Official Review · Reviewer_beZD · 2025-10-29

**Soundness:** 2
**Presentation:** 3
**Contribution:** 1
**Rating:** 2
**Confidence:** 5

**Summary:**

The paper proposes LoMIME, a label-only membership inference attack that first extracts a surrogate model from a target API model and then performs membership inference offline. The core idea is to amortize query costs by combining query-efficient model extraction with unsupervised label-only MIA based on decision boundary distances. The authors claim this approach achieves competitive membership inference performance compared to prior label-only attacks, with fewer total queries.

**Strengths:**

- The paper provides a well-structured overview of prior work in label-only MIAs and model extraction, and positions its contribution as bridging the two under restrictive black-box assumptions.

- The paper is logically clear and easy to follow.

**Weaknesses:**

- The proposed attack pipeline can be trivially mitigated by standard model extraction defenses (API monitoring, or randomness injection in predictions). Since the approach depends entirely on extracting a high-fidelity surrogate, any defense that disrupts extraction would nullify the attack. The paper does not evaluate or even discuss these well-known countermeasures, undermining its practical relevance.

- Although the paper claims query efficiency, the total number of required queries remains substantial. In practice, privacy concerns are often case-specific, where an attacker is typically interested in testing the membership status of a small number of sensitive samples rather than auditing large populations. Therefore, the proposed amortization argument—that the one-time extraction cost can be justified when inferring many samples—appears weak and unrealistic.

- All experiments are performed on simple, overfitted feedforward networks trained on small tabular datasets. These settings are known to overestimate MIA success. No evaluation is conducted on more realistic architectures (e.g., ResNet) or larger-scale datasets (e.g., CIFAR-10), which significantly limits the generality of claims.

- The paper lacks a comparison with state-of-the-art label-only membership inference attacks, such as OSLO, in terms of both query efficiency and attack success rate.

- In practical MLaaS settings, API models are frequently updated. Since the surrogate extraction in LoMime is a one-time process, its relevance quickly diminishes if the target model changes. The paper neither analyzes nor discusses this inevitable limitation.


[r1] Peng, Yuefeng, et al. "Oslo: one-shot label-only membership inference attacks." Advances in Neural Information Processing Systems 37 (2024): 62310-62333.

**Questions:**

- How would the proposed attack perform under standard model extraction defenses?

- Why are all evaluations conducted only on small tabular datasets with simple feedforward networks? Can the method generalize to more realistic and complex architectures such as ResNet or datasets like CIFAR-10?

- How does the proposed approach compare in terms of both query efficiency and attack success rate with recent state-of-the-art label-only MIA methods (e.g., OSLO)?

- How would the proposed framework handle frequent model updates, which are common in real-world MLaaS systems? Does surrogate extraction remain effective or require full re-extraction each time?

---

### Official Review · Reviewer_Y1tL · 2025-10-31

**Soundness:** 2
**Presentation:** 3
**Contribution:** 2
**Rating:** 2
**Confidence:** 4

**Summary:**

This paper has a rough two stage attack:

First, the paper does model extraction: it extracts a surrogate model S from the black-box target M using query-efficient active learning (from MARICH) combined with synthetic data generation via perturbation (from AUTOLYCUS). This is a fairly low total query cost, and can be averaged over the full attack process.

Then, the paper has an offline MIA phase: it performs label-only membership inference on the surrogate S (using decision boundary distance) rather than on M. Since S is extracted, the paper can query it unlimited times offline at no cost to the target.

**Strengths:**

The paper presents a reasonable evaluation with ROC curves on three datasets that allow the reader to understand what the attack does.

**Weaknesses:**

My primary concern with this paper is that all of the evaluations are for ML models that are *severely* overfit. As the paper writes: "Note that all target models are overfitted and achieve close to perfect training accuracy (≈ 100%)". This is the wrong approach to take for MIA schemes: because we know that the primary driver of membership inference is overfitting, running an attack when the defenses are most vulnerable doesn't make sense (Carlini et al. 2022). I would be much more interested in a paper that introduces an MIA attack that works on strong baseline defenses. The argument that this is done "to simulate realistic privacy risks in deployed machine learning systems" doesn't convince me: most ML models in practice are *not* overfit.

This particularly concerns me because it looks like most of the ROC curves suggest that this is mostly a non-membership inference attack that is relying on exactly this over-fitting, and I do not believe it would work for non-overfit models.

Concretely, this also means I would like to see datasets that are frequently used in prior work that introduce strong attacks, like CIFAR-10 or CIFAR-100. "While these datasets ostensibly have privacy-relevance, we believe it is more important to study datasets that reveal interesting properties of machine learning than datasets that discuss privacy. We nevertheless present these results in the Appendix, but encourage future work to omit these results and focus on the more informative tasks we consider" (Carlini et al. 2022).

**Questions:**

Can you provide a 1-to-1 comparison to some prior attacks, both label-only and not, on your models?

Can you run your attack on models that aren't severely overfit?

I don't understand, on the method, why you can't just run a standard membership attack on S? Why do you need to run label-only attack on S? This doesn't make sense to me. You've just stolen S so clearly you have it.

---

### Official Review · Reviewer_wZPq · 2025-11-01

**Soundness:** 2
**Presentation:** 2
**Contribution:** 2
**Rating:** 2
**Confidence:** 5

**Summary:**

LOMIME is a label-only membership inference attack that first steals a surrogate model from a black-box target using query-efficient model extraction. After this one-time extraction, membership inference is done entirely offline on the surrogate, eliminating per-sample query costs. On benchmark datasets, the approach matches some early label-only attacks while using far fewer queries. Most standard defenses offer limited protection, except DP-SGD, which significantly reduces leakage.

**Strengths:**

1. Good writing, easy to follow.

2. The paper tries to solve an important problem.

**Weaknesses:**

1. **Incremental contribution**: While the paper presents a unified two-stage attack pipeline, much of its methodology is composed of components adapted from prior work. On the extraction side, the query-efficient active sampling strategy closely follows MARICH, while the synthetic-perturbation-based data generation is derived from AUTOLYCUS. On the inference side, the membership attack relies on standard label-only decision-boundary measurements similar to earlier works. As a result, the novelty lies primarily in integrating these techniques rather than introducing fundamentally new algorithmic ideas.

1. **Limited defense evaluation**: The paper evaluates only three classical defenses—DP-SGD, Dropout, and L2 regularization—and only on the Location dataset. Modern deployed APIs typically use a much broader defensive stack, including ensemble voting, randomized prediction masking, aggressive rate-limiting, query throttling, or even honey-labeling strategies designed specifically to foil extraction, e.g. PRADA [1]. By omitting these more realistic defenses, the paper substantially underestimates the practical variability and robustness of real-world deployments.

2. **Not necessarily query efficient**: Although amortization across many victim samples is used as motivation, the paper does not provide a quantitative cost-benefit analysis: how many membership queries must be amortized before extraction becomes worthwhile? More critically, large-scale extraction is highly conspicuous and vulnerable to rate-limiting or anomaly detection on commercial APIs. In practice, the method trades per-sample query cost for a massive front-loaded extraction phase, which only makes sense if the attacker needs to infer membership for thousands or millions of users—an unrealistic assumption for many threat models.

3. **Limited and unrealistic scenario**: All experiments are restricted to tabular datasets, which are arguably the simplest domain for label-only MIAs. No results are provided for images, text, or large language models, despite the fact that most real-world ML services operate in these modalities. The current evaluation, therefore, does not demonstrate scalability or generalizability.

4. **Missing important baseline**: Several recent label-only MIAs already achieve high accuracy with extremely low query budgets—including one-shot attacks that require only a single query per sample. Representative works include [2][3][4]. The absence of comparison against these state-of-the-art methods makes it impossible to judge whether the proposed extraction-first pipeline offers any real advantage. Without these baselines, the claimed contribution remains unsubstantiated.


## References
[1] Juuti, Mika, et al. "PRADA: protecting against DNN model stealing attacks." 2019 IEEE European Symposium on Security and Privacy (EuroS&P). IEEE, 2019.

[2] Peng, Yuefeng, et al. "Oslo: one-shot label-only membership inference attacks." Advances in Neural Information Processing Systems 37 (2024)

[3] Wu, Yutong, et al. "You only query once: An efficient label-only membership inference attack." The Twelfth International Conference on Learning Representations. 2024.

[4] Li, Hao, et al. "Enhanced {Label-Only} membership inference attacks with fewer queries." 34th USENIX Security Symposium (USENIX Security 25). 2025.

**Questions:**

see weakness

---

### Official Review · Reviewer_6tzR · 2025-11-06

**Soundness:** 2
**Presentation:** 2
**Contribution:** 1
**Rating:** 0
**Confidence:** 5

**Summary:**

This paper proposes a new label-only membership inference attack that first leverages model extraction to obtain a surrogate model that preserves the membership information of the target model, and then queries this surrogate for the actual attack. Experiments on three tabular datasets demonstrate the efficiency of the proposed method.

**Strengths:**

* The efficiency of label-only membership inference attacks is an important and practical research problem.

* The experimental results show that the proposed method is computationally efficient while achieving good attack performance.

**Weaknesses:**

* It is unclear why the main experiments are not compared with existing methods. Although the authors state on Page 8 that previous approaches are too costly in terms of queries, a comparison under the same setting would provide a clearer understanding of the trade-off between efficiency and effectiveness. Moreover, state-of-the-art MIA methods such as LiRA should also be included in the comparison.
* Several recent label-only MIA methods that focus on efficiency are not mentioned or compared, such as [1]. This omission makes the novelty and contribution of the paper unclear.
* The datasets used (tabular data) and the neural network architectures (very small MLP models) are not realistic for practical attack scenarios. As noted in the LiRA paper, such datasets have been largely abandoned. More relevant benchmarks such as CIFAR-10/CIFAR-100 with ResNet are preferred.


[1] Chameleon: Increasing Label-Only Membership Leakage with Adaptive Poisoning. ICLR 2024.

**Questions:**

* Why are no comparisons provided with prior MIA methods?

* Why are recent baselines not included in the evaluation?

* Why are only tiny tabular datasets and small MLPs  used in the experiments?

---

### Meta-Review · Area_Chair_5cQw · 2026-01-01

**Summary:**

Although all of the reviewers have positive scores, the authors did not address any of the concerns. The most significant concern is that there is no comparison with state-of-the-art MIA approaches provided, nor with any recent baselines in the evaluation. I recommend rejecting this paper.

**Reviewer Concerns:**

* Reviewer 6tzR: The authors did not address/respond to any of the reviewers' concerns.
* Reviewer wZPq: The authors did not address/respond to any of the reviewers' concerns.
* Reviewer Y1tL: The authors did not address/respond to any of the reviewers' concerns.
* Reviewer beZD: The authors did not address/respond to any of the reviewers' concerns.

**Reviewer Scores:**

* Reviewer 6tzR: I do not think the reviewer would have changed their score.
* Reviewer wZPq: I do not think the reviewer would have changed their score.
* Reviewer Y1tL: I do not think the reviewer would have changed their score.
* Reviewer beZD: I do not think the reviewer would have changed their score.

---

### Decision · Program_Chairs · 2026-01-26

Reject